# The Effect of Behind-The-Scenes Encounters and Interactive Presentations on the Welfare of Captive Servals (*Leptailurus serval*)

**DOI:** 10.3390/ani10040743

**Published:** 2020-04-24

**Authors:** Lydia K. Acaralp-Rehnberg, Grahame J. Coleman, Michael J. L. Magrath, Vicky Melfi, Kerry V. Fanson, Ian M. Bland

**Affiliations:** 1Animal Welfare Science Centre, Faculty of Veterinary and Agricultural Sciences, University of Melbourne, Parkville, Victoria 3052, Australia; grahame.coleman@unimelb.edu.au (G.J.C.); ibland@unimelb.edu.au (I.M.B.); 2Department of Wildlife Conservation and Science, Zoos Victoria, Parkville, Victoria 3052, Australia; mmagrath@zoo.org.au; 3Hartpury University, Gloucester GL193BE, UK; vicky.melfi@hartpury.ac.uk; 4Centre for Integrative Ecology, Deakin University, Geelong, Victoria 3216, Australia; kerryfanson@gmail.com

**Keywords:** human–animal interaction, live animal encounter programs, small felids, visitor effects, zoo animal welfare

## Abstract

**Simple Summary:**

Live animal encounter programs are an increasingly popular occurrence in the modern zoo. The effects of such encounters on program animal welfare have not been studied extensively to date. The aim of this study was, therefore, to explore animal welfare effects associated with encounter programs in a small felid, the serval, which is commonly involved as a program animal in zoos. Specifically, this study investigated how serval behaviour and adrenocortical activity (level of faecal cortisol metabolites) were affected by short-term variations in encounter frequency. Over the course of the study, the frequency of encounters was manipulated so that servals alternated between four different treatments, involving interactive presentations, behind-the-scenes encounters, both activities combined, or no interaction at all. The cats exhibited a significant reduction in stereotypic pacing on weeks when participating in interactive presentations, or the two activities combined. However, behavioural diversity (total number of behaviours exhibited) was strongly reduced on weeks when cats participated in both activities. Adrenocortical activity did not vary significantly between treatments. The reduction in stereotypic pacing suggests that involvement in an encounter program may exert a positive short-term welfare effect on the individual servals. A reduction in behavioural diversity, which was not considered a negative welfare effect in the short term, may, however, warrant some need for caution if a more frequent encounter program was to be implemented long-term. These findings contribute to the current knowledge of visitor–animal interaction in zoo-housed felids, which is very limited to date, and could also provide valuable guidance to zoo professionals that are currently engaging in an encounter program with servals or planning on implementing such a program in the future.

**Abstract:**

The serval (*Leptailurus serval*) is a small African felid that is well represented in zoos and often serves as an animal ambassador in encounter programs with zoo visitors. The impact on serval welfare in relation to such programs has not been investigated to date, and the aim of this study was to assess short-term welfare effects of varying levels of visitor interaction in two captive servals. Weekly blocks of four different treatments were imposed three times on each animal over 12 weeks, and the treatments involved (1) Presentations (serval undertaking a routine training session in a designated presentation space, typically attracting high visitor numbers), (2) Behind-the-scenes (BTS, a close encounter allowing a small group of visitors to interact closely with the cat in its enclosure), (3) Presentations and BTS combined, and (4) No visitor interaction. Serval activity budgets as well as behavioural diversity were created from behaviours observed from Close Circuit Television (CCTV) footage during four daily recording sessions per animal over three consecutive days per treatment, using instantaneous scan sampling every 60 s. Individual faecal samples were collected daily to monitor changes in faecal glucocorticoid metabolite (FGM) concentration. Results indicate that the mean number of scans with stereotypic pacing was significantly reduced (*p* = 0.01) during Treatments 1 and 3, when cats participated in presentations only, or the two activities combined. Conversely, a significant reduction in behavioural diversity (*p* < 0.001) was observed when cats participated in Treatment 3, i.e., cats expressed fewer behaviours when interaction with visitors was more frequent. FGM concentrations did not vary significantly with treatment (*p* > 0.05). Given the reduction in stereotypic pacing, these findings suggest that involvement in an encounter program appears to exert an overall positive short-term welfare effect on the individual servals in this study. Although a reduction in behavioural diversity was not considered a negative welfare effect in the short term, potential long-term negative welfare effects resulting from a more frequent encounter program could not be ruled out in the present study.

## 1. Introduction

The serval (*Leptailurus serval*) is a small felid that is native to various regions of sub-Saharan Africa [1,2,3,4]. The species is a wetland specialist that navigates between marshes and reed beds in search of its main food source, which is small rodents [1,2,3,4]. A characteristic hunting technique where the serval relies on its superior sense of hearing to locate prey in tall grasses, after which it then leaps and pounces on the prey item, makes it a highly successful predator [2,4]. Like most other felids, the serval is a solitary species [4]. The female commonly gives birth to a litter of 1–3 kittens in summer and cares for her young for a considerable amount of time after they become mobile [4].

The serval is currently classified as “least concern” [5], although wild populations have recently declined in parts of their range due to secondary poisoning by consumption of poisoned rodent prey [3]. Wild servals are also subjected to attacks by domestic dogs or are shot by local farmers to prevent predation on domestic poultry [6]. Fortunately, the species is well represented in captivity [7] and attempts at re-introducing captive individuals into the wild have proven successful [2]. The reproductive success rate among captive servals is, however, relatively low [8]. As with many small felids, very little is known about the welfare of servals in captivity, since most research efforts to date have been dedicated to the larger and more conspicuous felids, *Panthera* sp. [9,10]. Welfare-oriented research in these species has mainly focused on implementing and assessing the effects of various enrichment strategies [11,12,13,14], exhibit designs [15,16], and housing constellations [17,18]. Although this research has often led to measurable improvements in welfare, similar studies in small cats are comparatively scarce (however, see References [19,20,21]).

Additionally, only a handful of studies to date have addressed the topic of visitor–animal interaction and its welfare consequences for captive felids, even though these animals may be exposed to unfamiliar humans on a daily basis [22], and close visitor interaction with felids is becoming increasingly common in zoos worldwide [23]. As such, a range of interactive programs are now offered with various felids. A common feature is to grant visitors access to off-limit areas where they interact with and tong-feed a big cat, most commonly a lion or a tiger, through a protective barrier [24]. Encounters with small felids, including cheetahs, *Axinonyx jubatus*, and servals, are often more tactile in nature and commonly allow visitors to pat and have their photos taken with a cat, or engage in an interactive walk together with a cat and its keeper [24]. In addition, servals are commonly featured as animal ambassadors in serendipitous encounters or educational workshops in zoos [24].

Although encounter programs with captive felids are now commonplace in many zoos, Szokalski et al. [23] consider this a controversial practice, given the solitary and elusive nature of many felids. There is, however, not enough empirical data available to suggest that close visitor interaction may exert a negative welfare impact on participating animals. Szokalski et al. [23] studied the effects of interactive programs on the behaviour of captive lions, *Panthera leo*, and cheetahs at two Australian zoos and observed high levels of stereotypic pacing among lions prior to encounters. The encounters involved protected contact feeding, and it was suggested that food anticipation, rather than stress caused by close visitor interaction, may have been responsible for this effect. Cheetahs, on the other hand, frequently expressed signs of affiliative behaviour towards both visitors and keepers during interactive walks, suggesting that it may have been a positive welfare experience for these individuals [23]. Another Australian research team studied fluctuations in faecal glucocorticoid (FGM) concentration in tigers, *Panthera tigris*, participating in walks, interactive presentations, and guest photo opportunities at two different zoos, and found that program animals had higher overall concentration of FGMs compared to non-participating animals at one institution, but the opposite trend was observed at the second zoo [25]. The authors suggest that variation in the level of conditioning and familiarity with the public may have been the reason for this disparity. There were, however, no behavioural observations that could further strengthen this claim, since the main focus of this study was to validate a physiological assay [25].

Given their widespread occurrence in zoos and their popular role as program animals, studies investigating the welfare impacts of encounters with captive servals is clearly worthwhile, in order to support the continued involvement of this species in interactive programs. Such research would optimise the care and welfare of individual animals and contribute to our understanding of visitor–animal interaction in small exotic felids. The aim of the current study was to explore the overall impact of encounters on the behaviour, physiology, and potentially short-term welfare of individual zoo-housed servals. Specifically, the aim was to determine whether potential welfare impacts were affected by:
Encounter frequency,Type of encounter (behind-the-scenes encounter involving a small number of visitors and close visitor–animal proximity, versus an interactive presentation with higher numbers of visitors but lower visitor–animal proximity).


It was hypothesized that variation in encounter frequency, as well as type of encounter, would elicit changes in serval behaviour and physiology that could be indicative of a welfare impact, though it could not be predicted whether this impact would be positive or negative due to the paucity of information on this topic. To our knowledge, our study was the first of its kind to investigate a potential cause–effect relationship between visitor interaction and behavioural and physiological welfare measures in captive servals.

## 2. Materials and Methods

All animal procedures in the current study were approved by the Zoos Victoria Research and Animal Ethics Committee (ZV16003).

### 2.1. Study Animals—Housing and Husbandry Routine

Study animals included two adult servals housed at Werribee Open Range Zoo (WORZ), which is situated 35 km southwest of Melbourne, Victoria, Australia. The servals, named Nanki and Morilli, were both captive-bred females from the same litter. They were born at Mogo Zoo, New South Wales, Australia, in December 2008, and were relocated to WORZ in March 2009. Although parent-reared from birth, once they arrived at WORZ, they were bottle-fed by keepers while being introduced to solid foods and were subsequently weaned at around six months of age. Their current feeding regime included two meals per day: one morning and afternoon feed consisting of 2–3 mice or day-old chicks each, and a rabbit or chicken leg each. In addition, each serval received two small portions of diced red meat per day, which was usually given as a reward during training sessions or visitor interaction.

The servals were housed off public display and could only be seen by the visiting public during presentations or behind-the-scenes (BTS) encounters. The servals were housed solitarily and alternated between an open yard (Figure 1b) and three adjacent pens (Figure 1a). The keepers swapped the cats’ housing on a daily basis. The open yard had a total area of approximately 75 m^2^ and a ground cover of sand and mulch. The yard was interspersed with logs, elevated platforms, and climbing structures, had a covered nest with a straw bed, a drinking trough, and a designated area for visitor interaction (Figure 1b). The pens had a total combined area of approximately 36 m^2^ and a ground cover of synthetic turf mats and sand. The pens also had elevated shelves, burlap hammocks, cat tunnels, drinking troughs, and heated beds for overnight rest (Figure 1a).

### 2.2. Visitor Interaction Program

Shortly after the servals arrived at WORZ as young kittens, the keepers began conditioning the cats for becoming ambassador animals in an interactive program. The animals were taught to follow basic instructions such as recall and sitting on command and were given food rewards and verbal praise to reinforce such behaviours. At the time of the study, the servals had participated in the program for seven years, and were able to perform a complex series of behaviours on command, with the aim of highlighting some of the serval’s natural foraging techniques to visitors (i.e., beam walks to simulate capture of roosting birds in a tree, leaps and pounces to simulate capture of small rodents, and retrieving a meat reward out of a pond to simulate capture of fish and frogs).

The current program involves a daily presentation and a BTS encounter that takes place four days a week. The servals alternated in participating in the daily presentation, but both individuals usually participated in BTS, although one at a time, so when one cat interacted with visitors, the other was held in the off-limit area until the keeper swapped them over. The daily presentation was included in the general zoo admission and took place at 11:00 am in a designated presentation space adjacent to the serval enclosures. The presentation typically attracted large numbers of visitors (maximum 250 people), but the audience did not interact with the serval (i.e., no feeding, touching) and the front row was seated approximately 2 m away from the front of the stage. The participating serval was escorted on a leash by its keeper to the presentation space. Once secured inside the space, the serval was encouraged to undertake a routine training session in front of the audience. The serval was rewarded with red meat for its efforts. The presentation typically lasted for 10–15 min and involved an educational message from the keeper about serval biology, captive management, and conservation of African wildlife.

The behind-the-scenes encounter incurred an additional fee for visitors and took place at 1:30 pm every Tuesday, Thursday, Saturday, and Sunday, in a designated interaction space inside the serval yard (Figure 1b). A small group of visitors (maximum 6 people) were escorted by a zoo volunteer into the yard where they got to meet the keeper and the servals. Children had to be at least 8 years old to participate, and participants were to stay seated and follow the keeper’s instructions at all times. The encounter involved a training session that was typically longer and involved more spectacular leaps than the presentation. The visitors were also given the opportunity to interact with the serval up close, by having their photograph taken with a cat, gently stroke the cat on its back and chest if the cat allowed, and have the cat lick a small quantity of cream cheese from the visitor’s finger. The encounter typically lasted for 30–40 min and involved a similar educational message as seen in the presentation.

On days when servals did not participate in any visitor interaction, they typically received a one-on-one training session with their keeper inside the serval yard or were taken for a walk by their keepers in the off-limit areas.

### 2.3. Experimental Design

Using a repeated measures design, the present study imposed changes to the regular program by implementing weekly blocks of four different treatments. The treatments were designed with the aim of separating the effects of presentations and BTS, as visitor number and visitor–animal proximity was markedly different between the two. Hence, the treatments involved participation in either presentations or BTS, the two combined, or no involvement in any visitor interaction (Table 1). Treatments were imposed for seven consecutive days and changed over every Friday of the week. Each study animal was subjected to each treatment for a total of three weeks, resulting in a total study period of 12 weeks. The two cats alternated between the four treatments, so that each animal was subjected to a different treatment each week. Treatments 1 and 2 always occurred together, i.e., when one cat undertook presentations, the other cat undertook BTS (Table 1). Likewise, treatments 3 and 4 always occurred together, so that one cat undertook both presentations and BTS while the other cat did not participate in any visitor interaction (Table 1). The reason for this was to cause minimal interference with the regular program, by ensuring that at least one cat would be available for either presentations or BTS in any given week. The order of the treatments was randomised prior to the onset of the study. Apart from changes to the level of visitor interaction, no other alterations to husbandry and housing were implemented during the study period. Keepers were encouraged to continue taking the cats for walks and undertake one-on-one training sessions when deemed necessary, so that the amount of training and physical activity remained relatively constant across treatments. The study was undertaken between May and September 2016.

### 2.4. Behavioural Observations

Qualitative behavioural assessments, where behaviours that could be indicative of both positive and negative welfare are monitored on an individual basis, is a widely used approach in welfare-related research [26,27,28,29]. In the present study, behaviour was monitored over the last three treatment days of each study week (Tuesday–Thursday) using cameras installed in the serval enclosures. A total of 14 cameras (PACOM Dome Close Circuit Television (CCTV)) were used, and they were positioned accordingly: 2–3 cameras in each serval pen, and seven cameras in the yard. The cameras were all connected to a DVR (PACOM Digital Video Recorder, Pacific Communications, Melbourne, Australia) and an external viewing source positioned in a crate in the nearby airlock. The software I-Watch (version 1.2.0, Shield Technology) was used for all subsequent video analysis. The cameras were set to record throughout each observation day. Four designated recording sessions were then sub-sampled from the footage, with the aim of capturing behaviour immediately prior to, during, and after visitor interaction, as well as in the morning and afternoon (Table 2). During presentations and BTS, only the behaviour of the non-participating cat was monitored (Table 2), since the participating cat was either in the presentation space or participating in BTS at this time. Since BTS was of longer duration than the presentation (typically lasted for 30–40 min), behaviour of the non-participating cat was monitored for the first 15 min of the encounter only. For the purpose of analysis, the four sessions were collapsed into two daily observation blocks (Table 2).

Behaviour was monitored by instantaneous scan sampling every 60 s [30]. Behaviour was not recorded if the keeper was in the enclosure at the time of scanning, as their presence tended to affect the cats’ behaviour. A comprehensive ethogram based on previously published data [27] was used as a basis for scoring serval behaviour. The final ethogram used in this study (Table 3) contained four different behavioural categories: passive, active, maintenance, and abnormal repetitive behaviours. At each observation session, the total number of behaviours that the cat was engaged in was referred to as behavioural diversity.

### 2.5. Adrenocortical Activity

Short-term as well as prolonged exposure to a stressor is often reflected as a change in circulating glucocorticoids, hence, cortisol and its associated metabolites is usually the preferred hormone of choice when assessing physiological stress responses [27,32]. In the current study, adrenocortical activity was monitored throughout the study period by analysing variation in excreted levels of faecal glucocorticoid metabolites (FGMs). This technique has been validated previously in a variety of mammal species, including felids [33,34], and is considered a reliable and non-invasive approach to measuring patterns of adrenocortical activity. Individually identifiable faecal samples were collected during each treatment day, and 1–2 days post treatment, to account for excretion lag time [35,36]. Typically, the cats defecated once a day, though occasionally somewhat more or less frequently. Each study day, keepers were instructed to collect all faeces from both individuals. As soon as the keeper in charge became aware that a cat had defecated, the keeper collected the entire scat, and placed it in a plastic zip-lock bag labelled with animal ID, date, and collection time. Samples could be anywhere between <1 h–15 h old upon collection (the latter applied if the cat(s) had defecated overnight and the sample was not collected until the next morning). Immediately upon collection, samples were transferred to a freezer (−20 °C) where they were stored until extraction and analysis. FGMs were extracted using the ethanol-vortex method [33,34] by adding 4 mL of 80% ethanol to 0.5 (±0.01) g of homogenised, wet faeces placed in 5 mL polypropylene vials. Capped vials were vortexed and placed on an orbital shaker overnight. The following day, samples were centrifuged for 5 min at 5000 rpm. The supernatant was then decanted into 1 mL microcentrifuge vials and assayed immediately. The total number of samples analysed per treatment was as follows: Treatment 1 (Presentations only)—23 samples, Treatment 2 (BTS only)—22 samples, Treatment 3 (Presentations + BTS)—27 samples, Treatment 4 (No interaction)—23 samples. Based on these samples, a mean value was generated for each treatment, in order to compare potential changes in FGM across treatments.

FGMs were measured using a group-specific glucocorticoid enzyme immunoassay that had previously been validated for felids in general [36,37] and servals in particular [38]. The corticosterone antibody and corresponding horseradish peroxidase conjugate were both obtained from J. Brown (Smithsonian Institution, Washington, DC, USA; Lab code Cs6). Assay protocols followed previously described methods [36]. Briefly, 96-well microtitre plates were coated with 150 µL of goat anti-rabbit IgG (2 μg/mL). Immediately prior to use, plates were washed three times. 50 µL of standard, control, or diluted faecal extract were added to each well, immediately followed by 50 µL of horse-radish peroxidase-conjugate (1:80,000) and antibody (1:100,000). The plate was then incubated for two hours at room temperature while shaking, then washed four times to remove unbound steroids. Subsequently, 150 µL of substrate solution was added to each well, and the plate was then incubated at room temperature for approximately 45 min, until the optical density of the maximum binding wells was >0.7. Optical density was read immediately on an Anthos 2010 plate reader (Anthos Labtec Instruments, Austria) at a wavelength of 450 nm. All samples were assayed in duplicate. FGM concentrations (expressed as ng/g wet faecal weight) were calculated from the standard curve using Skanlt RE 4.1 software. To monitor precision and reproducibility, low (∼70% binding) and high (∼30% binding) control samples were run on each plate (four plates were run in total). Inter-assay coefficients of variation (CVs) for low and high controls were 2.6% and 4.5%, respectively. Intra-assay CV was <15%. The assay was biochemically validated by demonstrating parallelism between a serially diluted sample pool and the standard curve.

### 2.6. Statistical Analysis

The data obtained in this study was analysed using GenStat version 16. To determine the effect of treatment on behaviour (Table 4) and adrenocortical activity (Section 2.5), a general analysis of variance was used, where individual animal, treatment (1—Presentation, 2—Behind the scenes, 3—Presentations + BTS, 4—No interaction) and session (observation block 1 or 2; Table 3) were the treatment structure (Table 1), and mean number of scans spent in passive, active, maintenance, and abnormal repetitive behaviours as well as concentrations of faecal glucocorticoid metabolites were the Y-variates. Additionally, when examining the effect of housing on behaviour, treatment and housing (pens or yard) were used as independent variables, and the above-mentioned behavioural measures were used as dependent variables in a general analysis of variance. In the analysis, degrees of freedom (d.f.) were: Treatment—3, Individual—1, Session—1, and Total d.f.—15.

In all analyses, a post-hoc Fisher’s least significant difference (LSD) was conducted to determine significance of differences between means for the independent variables. A significance level of 0.05 was used in all analyses.

## 3. Results

### 3.1. Passive, Active, and Maintenance Behaviours

The cats spent, on average, 55–75% of their time in passive behaviours. Sitting, standing, and resting awake were all frequently observed, whereas sleeping and hiding were seen more sporadically. Total time spent in passive behaviours, as well as time spent sleeping, hiding, playing, and investigating, did not differ significantly between treatments (Table 4). There was, however, a significant treatment effect of time spent sitting (*p* = 0.05), since cats spent less time in this behaviour during BTS (Table 4).

Active behaviours and maintenance behaviours typically occupied between 5% and 15% of the cats’ time. Walking was the most commonly observed activity, as the cats were frequently moving around within the enclosure or patrolling the enclosure perimeter. Similarly, eating was the most frequently observed maintenance behaviour that contributed most to overall maintenance levels. Significantly higher levels of running, climbing, and jumping (these three behaviours were observed in low frequencies and were therefore combined for the purpose of analysis) were observed in the BTS treatment (*p* = 0.01; Table 4). The combined treatment (BTS + Presentations) appears to have induced overall lower activity levels, since total time spent in active behaviours, as well as time spent walking, was significantly reduced in this treatment (*p* = 0.04 and 0.03, respectively; Table 4). Maintenance behaviours were highly consistent across treatments, apart from scent marking, which increased significantly when cats participated in Presentations only (*p* = 0.04; Table 4).

### 3.2. Abnormal Repetitive Behaviours

Stereotypic pacing was observed in both cats, who spent, on average, 15–25% of their time in this behaviour. Interestingly, treatment exerted a noticeable effect on pacing levels, as pacing increased significantly during No interaction and BTS treatment (*p* = 0.01; Figure 2a, Table 4). A highly significant interaction between treatment and time of day (*p* < 0.001) was also discovered in regard to pacing, as the cats spent more time pacing in the latter half of the day during Presentations, but the opposite trend was observed in BTS treatment (Figure 2b). No such effect was observed in Presentations + BTS or No interaction, though (Figure 2b). This difference was likely due to the timing of the daily visitor interaction. During presentations, cats participated in the morning presentation, but were excluded from the afternoon BTS. The pacing was then clustered in the latter half of the day (Figure 2b), when the other cat was undertaking BTS. Likewise, in BTS treatment, cats participated in the afternoon BTS but not in morning presentations. A surge in pacing was then observed in the first half of the day (Figure 2b), when the other cat was in the presentation space. In Presentations + BTS, cats participated in both activities, hence no peak in pacing behaviour was observed in either the morning or afternoon (Figure 2b). In No interaction, cats did not participate in any visitor interaction, hence pacing peaked both in the morning and afternoon, which led to overall higher levels of pacing during this treatment, but no effect of day (Figure 2a,b).

### 3.3. Behavioural Diversity

Behavioural diversity was shown to decrease markedly in Presentations + BTS compared to other treatments (*p* < 0.001; Figure 3; Table 4), i.e., the cats undertook fewer behaviours when visitor interaction was more frequent. Similar to pacing, a highly significant interaction (*p* < 0.001) between treatment and time of day was also observed, as behavioural diversity increased in the latter half of the day in Presentations, but the opposite was true for BTS treatment (Figure 3b). No such effect was observed in Presentations + BTS or No interaction (Figure 3b). This difference was most likely related to the activity the cat was currently participating in. Higher behavioural diversity in the latter half of the day in Presentations (when the cat was participating in morning presentations) versus the morning in BTS treatment (when the cat was participating in afternoon BTS), suggests that whenever there was an absence of visitor interaction, this led to an engagement in a wider variety of behaviours. This result is also consistent with behavioural diversity being overall lower in treatment 3, the treatment where the cats participated in visitor interaction in both the morning and afternoon (Figure 3).

### 3.4. Effect of Individual and Time of Day on Behaviour

Although the two cats responded similarly to treatment, there were some noticeable individual differences in time budgets, and time of day exerted some effects on pacing and maintenance behaviours, with pacing being significantly higher in the latter half of the day (0.04), presumably due to food anticipation which often coincided with the afternoon observations. Eating (*p* < 0.001), grooming (*p* = 0.02), and total time spent in maintenance behaviours (*p* < 0.001) were also significantly elevated in the latter half of the day.

### 3.5. Adrenocortical Activity

Concentrations of FGM did not vary significantly between any of the treatments (overall mean 400.25 ng/g). Hence, there is insufficient evidence that variation in visitor interaction affected adrenocortical output in the servals in the present study.

## 4. Discussion

### 4.1. Behavioural Measures

Outside of visitor interaction, cats spent the majority of their time in passive behaviours, and a relatively short amount of time in active or maintenance behaviours. This is consistent with findings described by other authors, in zoo-housed servals [39] as well as many other felids [20,40,41,42,43,44,45]. Hence, a daily pattern involving extended periods of rest interspersed with short bursts of activity appears to be a recurring phenomenon amongst captive felids.

Overall levels of inactivity did not differ significantly across treatments (Table 4). However, the cats appear to have been more active while participating in behind-the-scenes encounters only, as shown by a reduction in time spent sitting (Table 4). Activity levels also surged during this treatment, during high energy activities (including running, jumping, and climbing) in particular (Table 4). Interestingly, this effect was only seen in this treatment, and not in the treatment involving both BTS and presentations. If the increased vigilance and activity was due to close-up interaction with visitors, one would have expected to see a similar trend during this treatment as well, but in fact, activity levels were significantly reduced when cats were involved in BTS + Presentations (Table 4). Hence, close-up visitor interaction per se is unlikely to be responsible for the difference observed, and further investigation would be necessary to explain these effects.

Although not commonly observed in this study, the fact that hiding was consistent across treatments is a positive finding. Frequent hiding has been positively correlated with high levels of urinary cortisol in leopard cats [46] and behavioural stress scores in domestic cats in confinement [47]. Hiding may also serve as a coping strategy while cats are exposed to a potentially stressful situation [48,49]. The fact that hiding was rarely observed, and did not differ with any of the treatments, is undoubtedly important from a welfare perspective.

Another behaviour that often serves as a welfare indicator is the incidence of stereotypic pacing. Pacing is a very common occurrence among captive felids [50,51,52,53] and was frequently observed in the present study. Interestingly, some noticeable treatment effects were identified in relation to pacing. Pacing significantly increased when the cats were excluded from visitor interaction, or when participating in behind-the-scenes encounters only (Figure 2a). A clear time-of-day effect was also seen when participating in BTS or presentations only, as pacing was clustered around the time when the cat was not engaging in these activities, i.e., when the other cat was undertaking either activity, there was a clear spike in pacing for the non-participating cat (Figure 2b). The servals would typically pace in anticipation as the keeper approached to collect a cat for presentations or BTS. Although pacing per se is not considered positive, anticipatory pacing may, in some cases, be indicative of positive welfare, since anticipatory behaviour is associated with the release of dopamine, which is linked to the expectation of rewards [54,55]. The cat who was not selected at the time often continued to pace until the conclusion of the activity, which is indicative of a certain level of restlessness, frustration, or boredom when excluded from participation. The pacing is unlikely to have resulted from separation anxiety in the absence of the other cat, since the cats were housed solitarily and typically avoided interaction or showed signs of aggression (i.e., hissing) when they encountered each other through the exhibit fence.

Clearly, both individuals appeared highly motivated to participate in interactive activities, judging from the anticipatory pacing and the extended pacing that resulted when excluded from such an opportunity. The absence of presentations appears to have exerted a somewhat stronger effect, since overall pacing levels were higher during BTS and No interaction. Broom [56] emphasises that when an animal shows a strong motivation to acquire a certain resource or action, it is most likely something that benefits the animal. In other words, an animal’s preferences can help us identify what could lead to improvements in welfare as the choice often seem to optimise welfare [57]. The results from the current study may therefore suggest that involvement in an interactive program exerts a positive effect on serval welfare, as they appear highly motivated to participate. Hence, visitor interaction may hold some enrichment potential for the servals in this study, and as such may be a fruitful strategy for optimising welfare for these individuals.

The overall level of pacing that the cats engaged in throughout the day averaged around 15–25%. This is on the high end compared to other studies in zoo-housed felids, which typically report daily pacing levels of around 5–15% [16,43,44,58]. Our numbers are, however, likely to be somewhat exaggerated as behavioural observations were undertaken mainly around the time of visitor interaction, when pacing tended to peak, thus may not be entirely representative of daily pacing levels. Still, enrichment strategies that elicit natural hunting and foraging behaviour, as well as introducing an element of novelty and unpredictability, may be an efficient tool in counteracting the overall incidence of stereotypic pacing, as previous studies in servals and other captive felids [39,41,58] have shown.

Behavioural diversity was strongly reduced on weeks when cats engaged in both presentations and BTS, and a strong time-of-day effect was also observed (Figure 3). When cats engaged in presentations or BTS only, behavioural diversity was reduced around the time of day when visitor interaction occurred (Figure 3b). Rehnberg et al. [47] identified a strong negative correlation between behavioural diversity and behavioural stress scores in domestic cats that were recently admitted to a new environment, with individuals experiencing higher stress load typically engaging in very few behaviours. In the present study, a more plausible explanation is that the cognitive and physical stimulation resulting from recent visitor interaction may have caused the cats to temporarily slow down and engage in fewer behaviours. A strong reduction in behavioural diversity while engaged in frequent visitor interaction may, however, warrant some caution. The combined treatment exposed the cats to a higher level of interaction than what they would normally experience in the regular program, and to avoid the risk of potentially overworking the cats, a more frequent activity schedule would need to be evaluated carefully if it was to be implemented on a permanent basis as potentially negative long-term welfare effects could not be ruled out in the present study.

### 4.2. Adrenocortical Activity

The current study found no evidence that adrenocortical activity was affected by short-term variation in encounter frequency. The FGM values obtained in this study were similar to those reported elsewhere in the literature on felids [25,36,47,59], suggesting that this technique may be a useful tool for monitoring physiological stress load in servals as well. It is, however, important to acknowledge the importance of sample size and treatment duration when attempting to measure an adrenocortical response. Also, the absence of a hormonal response does not necessarily mean an absence of a stressor [60]. Other biomarkers used in the assessment of short-term changes in adrenocortical activity [61,62] may offer additional insight and would be encouraged in future studies.

### 4.3. Other Influences

The present study identified a number of treatment effects on behaviour. However, it is difficult to determine if visitor interaction per se was responsible for the effects observed. While the servals showed a strong interest in participating in both presentations and BTS, it remains unclear whether the underlying motivation was related to visitor interaction or something else, i.e., palatable food rewards, cognitive stimulation, interaction with a familiar caretaker, or a temporary change in environment. The latter was only applicable to presentations, and thus could perhaps explain why the servals appeared particularly motivated to participate in this activity. To ascertain the actual influence of visitors, the study could be replicated in the presence of familiar keepers only during presentations and BTS. If similar results were obtained, one could conclude that the opportunity to interact with visitors may not be the primary motivator for the servals. Nevertheless, if the cats found visitor interaction highly aversive, the motivation to participate would most likely be diminished and one would expect to see some negative welfare indicators in relation to these activities. It is therefore unlikely that visitors exerted a negative influence on the cats’ welfare, although it remains unclear whether visitors were a positive or merely a neutral influence for the cats.

The provision of palatable food rewards is likely to have influenced the servals’ motivation to participate. Szokalski et al. [23] identified higher levels of anticipatory pacing in lions on days involving interactive tours as opposed to non-tour days, supposedly because visitors fed the lions during the tours. It is therefore plausible that the lions as well as the servals in the present study have learnt to associate visitor activities with food rewards [63,64]. The servals typically received diced red meat as a reward for carrying out a certain behaviour on cue during training sessions and encounters. Hence, it is possible that they were attracted not only to the food rewards but the activity of acquiring it as well, since it has been demonstrated in multiple species that animals may choose to work for food rather than obtaining it for free, a phenomenon known as contra-freeloading [65,66,67,68]. Cognitive challenges are thought to be an important contributor to positive welfare amongst zoo animals [69] and could be highly relevant to the serval, which is considered an active and intelligent species that requires opportunities for mental and physical stimulation in captivity [52].

Another factor that possibly attracted the servals is the opportunity to spend time with a familiar caretaker. Both servals frequently initiated affiliative interactions (including for example leg rubbing and ‘head butting’) towards their primary keepers, in a similar manner to what has been described between domestic pet cats and their owners [70]. Similar findings have been reported in captive cheetahs [23], who expressed affiliative behaviour towards both keepers and visitors during interactive walks. Indeed, the servals appeared to be closely bonded to their two primary keepers, who had been working closely with the cats for several years, and one of the keepers was responsible for rearing them when they first arrived to WORZ as young kittens. Hand-rearing may have increased their overall tolerance for close contact with people [63,71], and hence rendered them more suitable for interactive activities. Szokalski’s [23] work on cheetahs also involved hand-reared animals, and so did Narayan’s [25] study on tigers at Australia Zoo that demonstrated a reduction in faecal glucocorticoids when participating in a similar program. It has been suggested that a positive keeper–animal relationship might be an important mediator in zoo animals’ responses to visitors [72,73]. This is a relatively unexplored research topic that warrants further investigation.

The fact that this study relied on only two test subjects limits the generalisations that can be drawn from these results. Hence, the suggested recommendations for management are mainly applicable to the individual study animals. Nevertheless, the methodology for assessing short-term welfare in relation to encounter programs should come in useful for future studies of servals and other zoo-housed felids participating in similar programs. The results from the current study will hopefully provide an incentive for further research in this field, since it has been demonstrated clearly that even short-term alterations in visitor interaction may exert some noticeable effects on behaviour and welfare.

## 5. Conclusions

The following conclusions can be drawn from the present study:
The current study found no evidence of a negative welfare impact in relation to participation in interactive presentations or behind-the-scenes encounters, at either a behavioural or physiological level. In fact, participation in these activities was shown to significantly reduce the incidence of pacing, suggesting that visitor interaction may contribute to positive welfare for the servals.A significant reduction in behavioural diversity was observed on weeks when cats participated in both presentations and BTS. Although not considered a negative welfare impact in the short term, one cannot disregard potential long-term negative effects of a more intense activity schedule, hence careful monitoring would be advised if the level of interaction was to be increased on a more permanent basis.It could not be determined whether visitor interaction per se or some other factors (i.e., food rewards, cognitive stimulation, etc.) was responsible for the effects observed, but for management purposes, the current findings are sufficient to lend support to the continued involvement of the servals in the encounter program, at the level of interaction that the cats normally experience.


## Figures and Tables

**Figure 1 animals-10-00743-f001:**
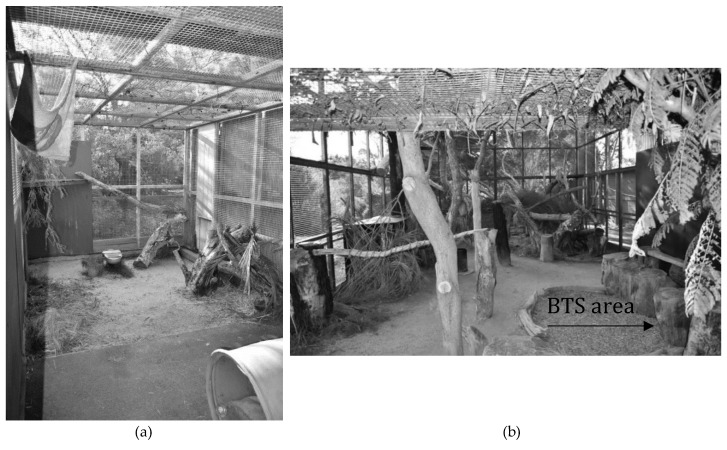
(**a**) Serval pen. The cats could move between the three adjacent pens via hatches that could be closed from the outside by the keeper when necessary. Pens were identical in size and had similar environmental features to the pen shown in the picture. (**b**) Serval yard with a designated area for behind-the-scenes (BTS) encounters on the right.

**Figure 2 animals-10-00743-f002:**
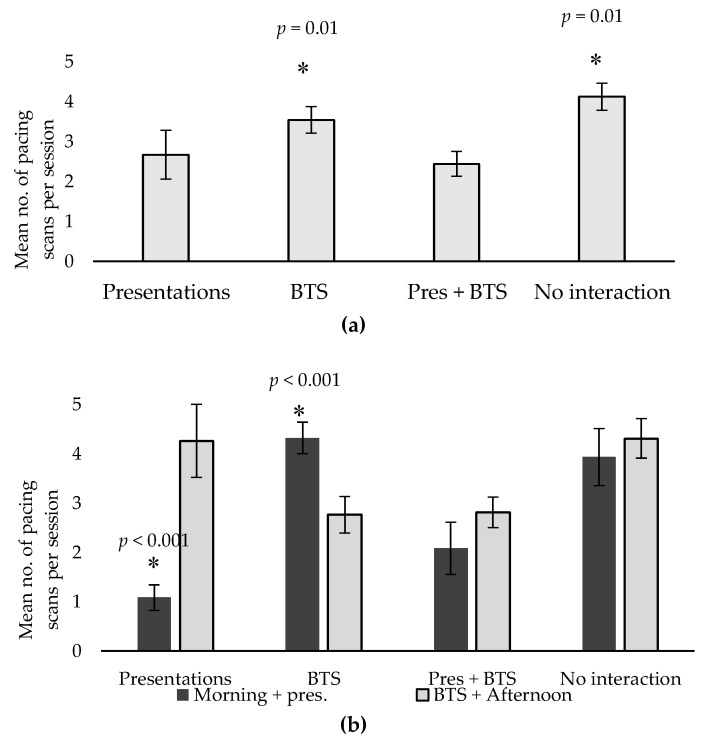
(**a**) Mean number of scans where pacing was observed during any given observation session across the four treatments. Error bars show standard error. Asterisks denote significant treatment differences. (**b**) Mean number of scans where pacing was observed during the two different observation blocks across the four different treatments. Pres–Presentations; BTS–Behind the Scenes Error bars show standard error. Asterisks denote significant treatment/session interactions.

**Figure 3 animals-10-00743-f003:**
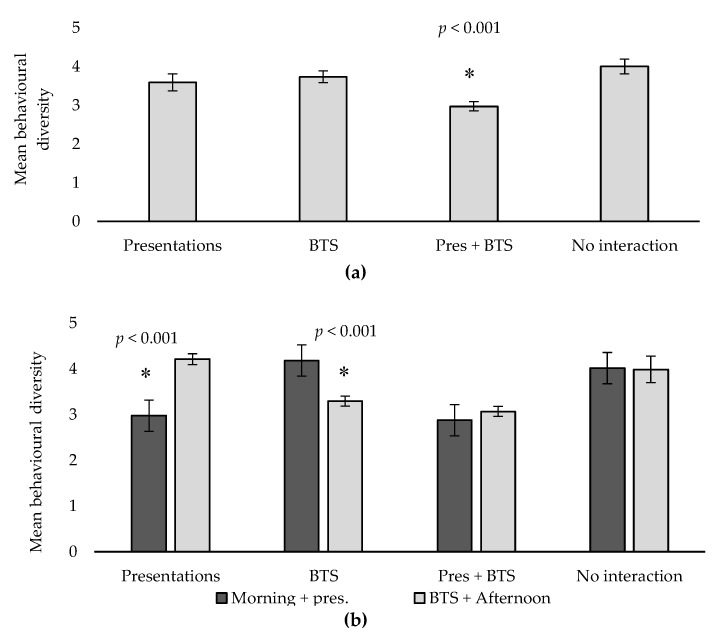
(**a**) Mean behavioural diversity (total number of behaviours expressed) during any given observation session across the four treatments. Error bars show standard error. Asterisks denote significant treatment differences. (**b**) Mean behavioural diversity during the two different observation blocks across the four treatments. Pres–Presentations; BTS–Behind the Scenes. Error bars show standard error. Asterisks denote significant treatment/session interactions.

**Table 1 animals-10-00743-t001:** Treatments imposed during the study period.

Treatment	Presentations	Behind-the-Scenes (BTS)
1—Presentation.	Yes	No
2—BTS	No	Yes
3—Presentation. + BTS	Yes	Yes
4—No interaction	No	No

Treatments 1 and 2 always occurred together throughout the study, so that one cat undertook presentations while the other undertook BTS. Likewise, Treatments 3 and 4 always occurred together throughout the study, so that one cat undertook presentations and BTS while the other did not participate in any visitor interaction.

**Table 2 animals-10-00743-t002:** Timeline of behavioural observations.

Session	Block	Time of Day	Duration	Total Observation Time Per Animal
Morning	1	9:00–9:30	30 min	30 min for both cats
Presentation	1	10:45–11:30 *	45 min	45 min for the non-participating cat, 30 min for the participating cat
BTS	2	13:15–14:15 **	60 min	45 min for the non-participating cat, 30 min for the participating cat
Afternoon	2	16:00–16:30	30 min	30 min for both cats

* Observations were undertaken 15 min prior to, during, and after presentation. ** Observations were undertaken 15 min prior to, first 15 min during, and 15 min after BTS.

**Table 3 animals-10-00743-t003:** Ethogram of serval behaviour, adapted from Stanton et al. [31].

Passive Behaviours	Active Behaviours	Maintenance Behaviours	Abnormal Repetitive Behaviours
**Sitting:** Cat is in an upright position, with the hind legs flexed and resting on the ground, while front legs are extended and straight, or crouching on top of all fours. **Resting awake:** Cat is lying down with its head raised and eyes open. **Sleeping:** Cat is lying down with its head down and eyes closed, performing minimal head or leg movement, and is not easily disturbed. **Standing**: Cat is in an upright position and immobile, with all four paws on the ground and legs extended, supporting the body. **Hidden:** Cat is fully concealed behind dense vegetation or burlap drape, to the point where it cannot be determined confidently which other behaviour(s) it is currently engaged in.	**Walking:** Forward locomotion at a slow gait. **Running:** Forward locomotion in a rapid gait, which is faster than walking or trotting. **Investigate:** Cat moves around attentively while sniffing the ground and/or objects or shows attention towards a specific stimulus by sniffing and/or pawing at it. **Climbing:** Cat ascends and/or descends an object or structure. **Jumping:** Cat leaps from one point to another, either vertically or horizontally. **Playing:** Cat interacts with and manipulates an object in a “playful” manner.	**Eat:** Cat ingests food (or other edible substances) by means of chewing with the teeth and swallowing. **Groom:** Cat cleans itself by licking, scratching, biting or chewing the fur on its body. May also include the licking of a front paw and wiping it over one’s head. **Defecate:** Cat releases faeces on the ground while in a squatting position. **Urinate:** Cat releases urine on the ground while in a squatting position. **Clawing:** Cat drags front claws along an object or surface. **Scratching:** Cat scratches its body using the claws of its hind feet. **Stretching:** Cat extends its forelegs while curving its back inwards. **Scent mark:** While standing with tail raised vertically, cat releases a jet of urine backwards against a vertical surface or object. The tail may quiver as urine is discharged. **Yawn:** Cat opens its mouth widely while inhaling, then closes mouth while exhaling deeply.	**Pacing:** Cat walks or runs back and forth in a repetitive manner along a designated path, without obvious purpose or intention. The cat had to traverse the same path at least twice to be considered pacing.

**Table 4 animals-10-00743-t004:** Effect of treatment on serval behaviour.

	Presentations	BTS	Presentations + BTS	No Inter-Action	Standard Error of Difference	*p*-Value
**Passive behaviours** *						
Total inactivity	10.58	9.88	10.17	11.38	0.75	ns
Sitting	5.20 ^bc^	3.74 ^ab^	4.58 ^b^	5.02 ^bc^	0.53	0.05
Standing	1.75	2.22	1.55	1.99	0.25	0.06
Resting awake	1.97	2.78	2.46	2.65	0.60	ns
Sleeping	0.40	0.02	0.15	0.39	0.19	ns
Hidden	1.26	1.11	1.43	1.33	0.59	ns
**Active behaviours** *						
Total activity	1.80 ^b^	2.22 ^ab^	1.45 ^bc^	1.84 ^b^	0.25	0.04
Walking	1.28 ^b^	1.51 ^ab^	0.96 ^bc^	1.40 ^ab^	0.18	0.03
Investigating	0.31	0.33	0.21	0.31	0.08	ns
Playing	0.06	0.11	0.12	0.02	0.07	ns
Other active (running, climbing, jumping)	0.16 ^a^	0.26 ^b^	0.16 ^a^	0.11 ^a^	0.04	0.01
**Maintenance Behaviours** *						
Total maintenance	1.19	1.28	0.78	1.18	0.33	ns
Eating	0.74	0.85	0.57	0.72	0.29	ns
Grooming	0.25	0.26	0.14	0.33	0.07	0.06
Scent marking	0.13 ^ab^	0.06 ^b^	0.03 ^bc^	0.03 ^bc^	0.04	0.04
Other maintenance (urinating, defecating, clawing, scratching, stretching, yawning)	0.07	0.11	0.04	0.10	0.03	ns
**Pacing** *	2.67 ^a^	3.54 ^b^	2.44 ^a^	4.12 ^b^	0.45	0.01
**Behavioural diversity** **	3.59 ^a^	3.74 ^a^	2.97 ^b^	4.00 ^a^	0.20	<0.001

* Depicts mean number of scans for any given observation session, ** depicts mean number of behaviours undertaken during any given observation session. Values with different superscript letters differ significantly from each other for each behaviour.

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
