# Peer review of "The Effect of Behind-The-Scenes Encounters and Interactive Presentations on the Welfare of Captive Servals (Leptailurus serval)"

_animals, 2020, doi:10.3390/ani10040743_

Round 1

Reviewer 1 Report

You have addressed all my comments with your changes. This is an interesting paper and I look forward to seeing it in press!

Reviewer 2 Report

The previous review comments were all addressed accordingly.

This manuscript is a resubmission of an earlier submission. The following is a list of the peer review reports and author responses from that submission.

Round 1

Reviewer 1 Report

Overall, this paper covers an interesting and worthwhile topic, one that is certainly in need of investigation. It is quite clear and well written, and the study design maximizes the amount of data that can be obtained from only 2 animals. Although I recognize that such small sample sizes are the normal for zoo animal studies, it would be nice to acknowledge the limitations of basing management decisions for the species as a whole on a study that only included two individuals; for instance, would it be possible to replicate this in other zoos? 

My specific comments follow, divided by paper section: 

Methods 

  • Line 186: It is my understanding that during normal zoo operations, both servals participate in BTS simultaneously (I.e. both cats were present together, at the same time in the same space). If this is true, it seems like normal BTS includes both increased visitor interaction (petting, etc.) and increased conspecific interaction, however you mention that for the trial only one serval participated in BTS at a time. Do you believe your results permit extrapolation to all BTS-type encounters at this zoo given that you excluded conspecific interaction from your testing? (I understand the reason you did so given the logistics of the situation). 
  • Line 245: I understand why you would exclude the time period during which the cat was participating in BTS/presentations (as their behaviour would be different by virtue of the tasks they were being asked to perform), but why did you monitor the behaviour of the non-participating cat during this time? I believe an explanation is warranted here. 
  • Line 256: Is there any previous research that suggests than scan sampling at 60 s intervals is appropriate, or was this decided based on logistics?  
  • Line 266: Is it expected that a change in FGM as a result of the day’s activities would be detectable in that day’s feces? What happened when the cat did not defecate on that day? Was there no FGM measurement to correspond to that day’s activity? 

Results 

  • Although you mention that participating cats were not observed during BTS/presentations in the methods section, the wording here (“during BTS”) suggests that you were observing the participating cat during BTS itself. Could you perhaps rephrase these to make it more obvious that you were observing the behaviour of the cats during BTS treatment rather than during BTS itself? This only became clear after re-reading the paper a few times. 

Discussion 

  • Line 437: Could you propose an explanation for this effect, if not due to visitor interaction? Moreover, could increased vigilance be associated with poor welfare (e.g. being ‘on edge’ and unable to rest despite a desire to, feeling uneasy or unsafe)? 
  • Line 460: You noted an increase in pacing in the non-participating cat when their conspecific was participating in a visitor activity. Are there other possible explanations for this pacing behaviour, other than frustration from being excluded (e.g. boredom)?  
  • Line 484: It seems like you could confirm that elevated pacing was due to observations during times of visitor interaction by calculating pacing levels for the morning and afternoon sessions independently of the presentation and BTS sessions. Is the degree of pacing lower (and more in line with previous reports) when you only look at the morning and afternoon sessions?  

Author Response

We would like to thank this reviewer for the overall positive feedback and thoughtful comments on the manuscript. We have responded to the reviewer's comments and revised the manuscript to the best of our abilities. Please see below for our response to specific comments.

Many thanks, Lydia Acaralp-Rehnberg and co-authors. 

Overall, this paper covers an interesting and worthwhile topic, one that is certainly in need of investigation. It is quite clear and well written, and the study design maximizes the amount of data that can be obtained from only 2 animals. Although I recognize that such small sample sizes are the normal for zoo animal studies, it would be nice to acknowledge the limitations of basing management decisions for the species as a whole on a study that only included two individuals; for instance, would it be possible to replicate this in other zoos? 

We thank the reviewer for this comment. To make it clear to the reader that any management recommendations presented in this study do not apply to the entire species, we have added a paragraph at the end of the Discussion where this has been emphasised clearly, along with the potential of replicating the methodology used in the current study to servals and other animals participating in similar programs (please see line 630-637).

My specific comments follow, divided by paper section: 

Methods 

  • Line 186: It is my understanding that during normal zoo operations, both servals participate in BTS simultaneously (I.e. both cats were present together, at the same time in the same space). If this is true, it seems like normal BTS includes both increased visitor interaction (petting, etc.) and increased conspecific interaction, however you mention that for the trial only one serval participated in BTS at a time. Do you believe your results permit extrapolation to all BTS-type encounters at this zoo given that you excluded conspecific interaction from your testing? (I understand the reason you did so given the logistics of the situation). 

We thank the reviewer for this comment and we would like to clarify that under normal zoo operations, both cats did participate in the same session of BTS, however one at a time to avoid aggression between the two. Basically, while one cat was undertaking BTS, the other cat was held in an off-limit area until the keeper swapped them over. Hence, no conspecific interaction was present during normal zoo operations. To clarify this to the reader, we have made a brief amendment to section 2.2 that describes the regular visitor interaction program.

  • Line 245: I understand why you would exclude the time period during which the cat was participating in BTS/presentations (as their behaviour would be different by virtue of the tasks they were being asked to perform), but why did you monitor the behaviour of the non-participating cat during this time? I believe an explanation is warranted here. 

We thank the reviewer for this comment. Preliminary behavioural observations that took place prior to the onset of data collection indicated that pacing tended to increase in the non-participating cat while the other cat undertook presentations or BTS. Since this was considered important from a welfare point of view, we decided to monitor the behaviour of the non-participating cat during this time.

  • Line 256: Is there any previous research that suggests than scan sampling at 60 s intervals is appropriate, or was this decided based on logistics?  

To justify our decision to use instantaneous scan sampling for this study, we have included a reference after this statement on line 275. The sampling interval was decided based on logistics, since it was the shortest possible sampling interval that was deemed feasible given the extensive camera set-up which required the observer to monitor the cats over 14 different cameras. A shorter sampling interval, or continuous behavioural monitoring, would have made the analysis too time-consuming given the large number of cameras used in the study.

  • Line 266: Is it expected that a change in FGM as a result of the day’s activities would be detectable in that day’s feces? What happened when the cat did not defecate on that day? Was there no FGM measurement to correspond to that day’s activity? 

We thank the reviewer for this comment and we have clarified to the reader on line 296-297 that the FGM analysis accounted for excretion lag time by collecting samples on each treatment day as well as 1-2 days post treatment. Hence, a change in FGM as a result of the day’s activities would most likely show up 1-2 days after. However, we would like to emphasise that FGM analysis was not undertaken on a day specific basis, but rather a mean value was generated for each treatment, in order to compare potential changes in FGM across treatments (please see section 3.5). This has been clarified on line 311-312.  

Results 

  • Although you mention that participating cats were not observed during BTS/presentations in the methods section, the wording here (“during BTS”) suggests that you were observing the participating cat during BTS itself. Could you perhaps rephrase these to make it more obvious that you were observing the behaviour of the cats during BTS treatment rather than during BTS itself? This only became clear after re-reading the paper a few times. 

We apologise for the confusion and thank the reviewer for this comment. To improve readability, we have rephrased the wording in the Results section to clarify to the reader that we refer to the BTS treatment and not the actual BTS session.

Discussion 

  • Line 437: Could you propose an explanation for this effect, if not due to visitor interaction? Moreover, could increased vigilance be associated with poor welfare (e.g. being ‘on edge’ and unable to rest despite a desire to, feeling uneasy or unsafe)? 

We thank the reviewer for this thoughtful comment. Since the increased activity observed during BTS treatment cannot necessarily be attributed to vigilance, we have decided to omit the point around increased vigilance in the discussion. We have also decided to omit the statement about increased time spent standing, since this result was not statistically significant. As stated in the discussion, it is difficult to propose an explanation for the increased activity without conducting further research (please see line 504-514).  

  • Line 460: You noted an increase in pacing in the non-participating cat when their conspecific was participating in a visitor activity. Are there other possible explanations for this pacing behaviour, other than frustration from being excluded (e.g. boredom)?  

We thank the reviewer for another thoughtful comment and we agree that boredom could also have played a role in explaining the increased levels of pacing. This has been added as a possible explanation on line 535-536.

  • Line 484: It seems like you could confirm that elevated pacing was due to observations during times of visitor interaction by calculating pacing levels for the morning and afternoon sessions independently of the presentation and BTS sessions. Is the degree of pacing lower (and more in line with previous reports) when you only look at the morning and afternoon sessions?  

We thank the reviewer for this comment and it is possible that pacing levels would be more in line with other studies if looking at mornings and afternoons separately. However, due to limitations of the statistical model, it was not feasible to analyse each of the eight recording sessions separately (as we would have lost too many degrees of freedom if we had not combined the eight recording sessions into two blocks). For the purpose of analysis, the eight recording sessions were therefore combined into two blocks. When looking at descriptive statistics of each session separately, there was definitely a trend towards lower pacing levels in the morning and afternoon, although the afternoon session usually saw a slight increase in pacing, presumably due to food anticipation since the afternoon feed often coincided with this observation session. We have decided not to comment on this specifically in the manuscript, however we have changed the wording slightly to indicate that recorded levels of pacing were likely to have been exaggerated due to monitoring behaviour in response to visitor interaction, when pacing tended to peak (please see line 554). 

Reviewer 2 Report

Based on serial behavioural observations and faecal glucocorticoid metabolites, the presented manuscript aimed at evaluating aspects of an already existing zoo animal encounter program including two servals and its potential impact on animal welfare. This kind of studies is important and necessary. The manuscript is well written and interesting to read. However, the ms needs some additional effort – especially the statistical analysis as well as the presentation of the results warrant improvement.

What I miss in the introduction is that the authors do not explain why they chose to evaluate potential changes in behaviour and faecal GCM – indeed, it is evident if a reader is within the field of animal welfare, but it should be explained. Please consider that GCM are not equalled stress levels, as stated by the authors in the discussion (but please see MacDougall-Shackleton, S. A., Bonier, F., Romero, L. M., & Moore, I. T. (2019). Glucocorticoids and “stress” are not synonymous. Integrative Organismal Biology, 1(1), obz017.)

Lines 175 – 176: The table contains identical information for both individuals - please include the information in the text and skip the table or reduce the table to one row - it saves the reader to go over the exact same information twice.

Line 188: It would have been interesting to include the actual number of visitors per treatment? Did you coincidentally collect this data?

Line 242 - I watch software – please include company and version

Line 265; I suggest to change the subheading to: Faecal glucocorticoid metabolites (FCM) or HPA axis activity - it is the only “physiological measure” i.e. only one.

Line 305: which type of GLM was used?

In general, the statistics need to be described more thoroughly. In the current form, it is difficult to relate the presented results with the currently described statistics. What was tested how exactly?  Further, I suggest using a general linear mixed model (and included animal ID as a random effect) instead of a glm. Furthermore, I would include each observation session (shown in Table 3), i.e. 8 instead of 2 blocks, also given the strong interaction effect of treatment and time of the day. The dataset should have has enough observation points to do so without over-fitting.

Table 5.: Please improve the table description in a way that the reader actually knows what you are showing here. What is s.e.d? Standard error? Standard deviation? Please clarify. If you are giving means of cans for the given observation session, you also need to provide standard deviations for each behaviour. The superscript letters indicating significant differences are a bit confusing – please choose different combinations. It would also be helpful to mention how you calculated behavioural diversity in the text – now it is only a footnote, and the reader wonders about that already in the stats part.

Fig. 2a and b; what are the significant levels denoted by the asterisks? (same for fig 3)

Line 342: you have to write either p = 0.001 or p < 0.001 but P =<.001 does not work – please correct this for all occasions in the results section.

Line 423: please change subheading to: Faecal glucocorticoid metabolites (FCM) or HPA axis activity

Table 6 is somewhat confusing – there is one row containing the names of the two individuals and the “s.e.d” value (please clarify this issue here as well) – until there is no significant difference – I suggest skipping this table.

Line 508: I am sorry but I could not find fig 4.2a

Line 512-513: well written – this is an important aspect!

Line 517: ....on several behaviours.

Author Response

We would like to thank the reviewer for the overall positive feedback and helpful comments on our manuscript. We have now revised the manuscript to the best of our abilities and responded to the reviewer's comments below. 

Many thanks,

Lydia Acaralp-Rehnberg and co-authors

Based on serial behavioural observations and faecal glucocorticoid metabolites, the presented manuscript aimed at evaluating aspects of an already existing zoo animal encounter program including two servals and its potential impact on animal welfare. This kind of studies is important and necessary. The manuscript is well written and interesting to read. However, the ms needs some additional effort – especially the statistical analysis as well as the presentation of the results warrant improvement.

What I miss in the introduction is that the authors do not explain why they chose to evaluate potential changes in behaviour and faecal GCM – indeed, it is evident if a reader is within the field of animal welfare, but it should be explained. Please consider that GCM are not equalled stress levels, as stated by the authors in the discussion (but please see MacDougall-Shackleton, S. A., Bonier, F., Romero, L. M., & Moore, I. T. (2019). Glucocorticoids and “stress” are not synonymous. Integrative Organismal Biology1(1), obz017.)

We thank the reviewer for this comment. It was difficult to find a place within the Introduction where this could be incorporated smoothly, hence we decided to briefly explain our choice of behavioural and physiological measures at the beginning of Methods section 2.4 and 2.5.

Lines 175 – 176: The table contains identical information for both individuals - please include the information in the text and skip the table or reduce the table to one row - it saves the reader to go over the exact same information twice.

We thank the reviewer for this comment and in response, we have deleted Table 1 and included this information in text instead in Section 2.1.

Line 188: It would have been interesting to include the actual number of visitors per treatment? Did you coincidentally collect this data?

Yes, this data was collected but the analysis could not identify any relationship between visitor numbers during presentations and serval behaviour. Hence, we have decided not to present this data in the current manuscript.

Line 242 - I watch software – please include company and version

These details have been included, as requested by the reviewer.

Line 265; I suggest to change the subheading to: Faecal glucocorticoid metabolites (FCM) or HPA axis activity - it is the only “physiological measure” i.e. only one.

We thank the reviewer for this comment and we have changed this subheading to ‘HPA axis activity’, in accordance with the reviewer’s suggestion.

Line 305: which type of GLM was used?

In general, the statistics need to be described more thoroughly. In the current form, it is difficult to relate the presented results with the currently described statistics. What was tested how exactly?  Further, I suggest using a general linear mixed model (and included animal ID as a random effect) instead of a glm. Furthermore, I would include each observation session (shown in Table 3), i.e. 8 instead of 2 blocks, also given the strong interaction effect of treatment and time of the day. The dataset should have has enough observation points to do so without over-fitting.

We thank the reviewer for the comments on the statistics section of the manuscript and we have attempted to address the concerns. Indeed, our own thoughts during the development of the experimental design mirror the thinking of the reviewer. The initial project design was done in conjunction with the statistician within our group – we considered that given the limitations in animal numbers they weren’t truly random effects. When one animal was assigned to a treatment the other animal had to assigned to the alternative (for logistical reasons to allow zoo activities to continue) – hence our decision to include the animal as part of the fixed effects or treatment design in the analysis. The decision to combine observation sessions was made having followed the same reasoning the reviewer outlines here – we felt however that the combination approach increased the reliability of and provided a smoothing of the data by minimising the effects of random events like climate, logistical considerations etc.  We understand that different approaches to statistics are available and welcome further comment for the reviewer. If necessary, we are happy to re-run our analysis if this explanation is not sufficient to address the reviewer’s concerns.

We have now referred directly to the exact terminology within the statistical package in the revised section (please see line 348-357) to aid clarity to the reader and apologise for any confusion caused by the ambiguous terms used previously.

Table 5.: Please improve the table description in a way that the reader actually knows what you are showing here. What is s.e.d? Standard error? Standard deviation? Please clarify. If you are giving means of cans for the given observation session, you also need to provide standard deviations for each behaviour.

s.e.d. stands for “standard error of the difference” and is a standard error for assessing the difference between a pair of parameters (e.g. elements of a table of means). The analysis produces mean values and s.e.d for the group of means and we have used those values directly. As s.e.d. uses standard deviations of each mean as part of its calculation it simplifies the presentation of the data table in our opinion, rather than having to list each mean ± standard deviation. We believe this is a standard statistical practise but will define the abbreviation in the relevant tables.

Table 5 cont: The superscript letters indicating significant differences are a bit confusing – please choose different combinations. It would also be helpful to mention how you calculated behavioural diversity in the text – now it is only a footnote, and the reader wonders about that already in the stats part.

To clarify how we used superscript letters to indicate significant differences, please see line footnote Table 5, line 386. Calculations of behavioural diversity has been clearly defined in the Methods section 2.4, please see line 280-281.

Fig. 2a and b; what are the significant levels denoted by the asterisks? (same for fig 3)

Significance levels have been added to the Fig. 2 and 3 as requested.

Line 342: you have to write either p = 0.001 or p < 0.001 but P =<.001 does not work – please correct this for all occasions in the results section.

We apologise for the error and have corrected this for all occasions in the Results section.

Line 423: please change subheading to: Faecal glucocorticoid metabolites (FCM) or HPA axis activity

We thank the reviewer for this comment and we have changed this subheading to ‘HPA axis activity’, in accordance with the reviewer’s suggestion, both in the Results and Discussion.

Table 6 is somewhat confusing – there is one row containing the names of the two individuals and the “s.e.d” value (please clarify this issue here as well) – until there is no significant difference – I suggest skipping this table.

As per the reviewer’s suggestion, we have deleted this table and made some brief changes to the text in section 4.2.  

 Line 508: I am sorry but I could not find fig 4.2a

We thank the reviewer for drawing our attention to this, it is a typographical error. However, this entire sentence has now been deleted in section 4.2 (see previous comment).

Line 512-513: well written – this is an important aspect!

We thank the reviewer for this positive comment.

Line 517: ....on several behaviours.

This was actually not a typographical error, it was meant to be ‘serval behaviour’, but to avoid confusion we have omitted ‘serval’ from the sentence.